# Synthetic neuronal datasets for benchmarking directed functional connectivity metrics

João Rodrigues and Alexandre Andrade

Institute of Biophysics and Biomedical Engineering, Faculty of Sciences, University of Lisbon, Campo Grande, Lisbon, Portugal

## ABSTRACT

**Background.** Datasets consisting of synthetic neural data generated with quantifiable and controlled parameters are a valuable asset in the process of testing and validating directed functional connectivity metrics. Considering the recent debate in the neuroimaging community concerning the use of these metrics for fMRI data, synthetic datasets that emulate the BOLD signal dynamics have played a central role by supporting claims that argue in favor or against certain choices. Generative models often used in studies that simulate neuronal activity, with the aim of gaining insight into specific brain regions and functions, have different requirements from the generative models for benchmarking datasets. Even though the latter must be realistic, there is a tradeoff between realism and computational demand that needs to be contemplated and simulations that efficiently mimic the real behavior of single neurons or neuronal populations are preferred, instead of more cumbersome and marginally precise ones.

**Methods.** This work explores how simple generative models are able to produce neuronal datasets, for benchmarking purposes, that reflect the simulated effective connectivity and, how these can be used to obtain synthetic recordings of EEG and fMRI BOLD signals. The generative models covered here are AR processes, neural mass models consisting of linear and nonlinear stochastic differential equations and populations with thousands of spiking units. Forward models for EEG consist in the simple three-shell head model while the fMRI BOLD signal is modeled with the Balloon-Windkessel model or by convolution with a hemodynamic response function.

**Results.** The simulated datasets are tested for causality with the original spectral formulation for Granger causality. Modeled effective connectivity can be detected in the generated data for varying connection strengths and interaction delays.

**Discussion.** All generative models produce synthetic neuronal data with detectable causal effects although the relation between modeled and detected causality varies and less biophysically realistic models offer more control in causal relations such as modeled strength and frequency location.

Corresponding author
João Rodrigues,
jprodrigues@fc.ul.pt

## INTRODUCTION

The field of brain research that studies connectivity relies in an ever evolving array of methods that aim at finding directed or undirected connectivity links from recorded neuronal datasets which pose different challenges such as having nonlinear relations, non-stationary dynamics, low signal-to-noise ratio (SNR), linear mixes or slow dynamics. Therefore, most studies that introduce or discuss connectivity metrics use not only real but also simulated datasets for demonstration and validity purposes on a controlled environment. These simulated datasets are obtained through generative models that simulate the conception of neuronal activity, like local field potentials (LFPs), with a certain degree of realism and can be followed by forward biophysical modeling where LFPs are transformed into other neuronal recording datasets like electroencephalography (EEG), magnetoencephalography or functional magnetic resonance imaging (fMRI) blood-oxygen-level dependent (BOLD) signals, for example. With these controlled simulations, it is possible to isolate the effect of certain parameters and understand how directed functional connectivity metrics perform in a realistic range of values.

As far as directed functional connectivity is concerned, Granger causality (GC) based metrics have been in the center of several studies that discussed their plausibility in the analysis of fMRI BOLD datasets due to the inherent low SNR, slow dynamics compared to neuronal activity and confounding effects due to variable vascular latencies across brain regions (*Deshpande, Sathian & Hu, 2009*; *Smith et al., 2010*; *Valdes-Sosa et al., 2011*; *Seth, Chorley & Barnett, 2013*; *Rodrigues & Andrade, 2014*). Most of these studies used simulated datasets to support their claims with generative models such as networks of 'on–off' neurons (*Smith et al., 2010*), multivariate vector autoregressive (MVAR) processes with real LFPs (*Deshpande, Sathian & Hu, 2009*; *Rodrigues & Andrade, 2014*) and columns of thousands of spiking units (*Seth, Chorley & Barnett, 2013*) and BOLD forward models consisting of linear operations or more biophysically realistic models. Other problems such as the effect of volume conduction in directed connectivity has also been addressed with LFPs being modeled as sinusoids and the forward EEG model as a linear mix of different LFPs (*Kaminski & Blinowska, 2014*).

The objective of this work is to present and test the ability of popular generative and forward models to produce synthetic neuronal datasets with modeled causal effects, to be used as benchmarks.

## METHODS

In this section, we succinctly present the methodological framework used in this study from the modeling techniques to the connectivity metrics employed. Figure 1 depicts the generative models to simulate LFPs and the forward models for fMRI BOLD signals and EEG signals. The ensuing subsections expand on each technique separately by laying theoretical foundations, mentioning available software and explaining why these simulations have been important in the development of brain connectivity tools. Although we present the generative model definitions in their general terms for any given network, in this work simulations only encompass two nodes.

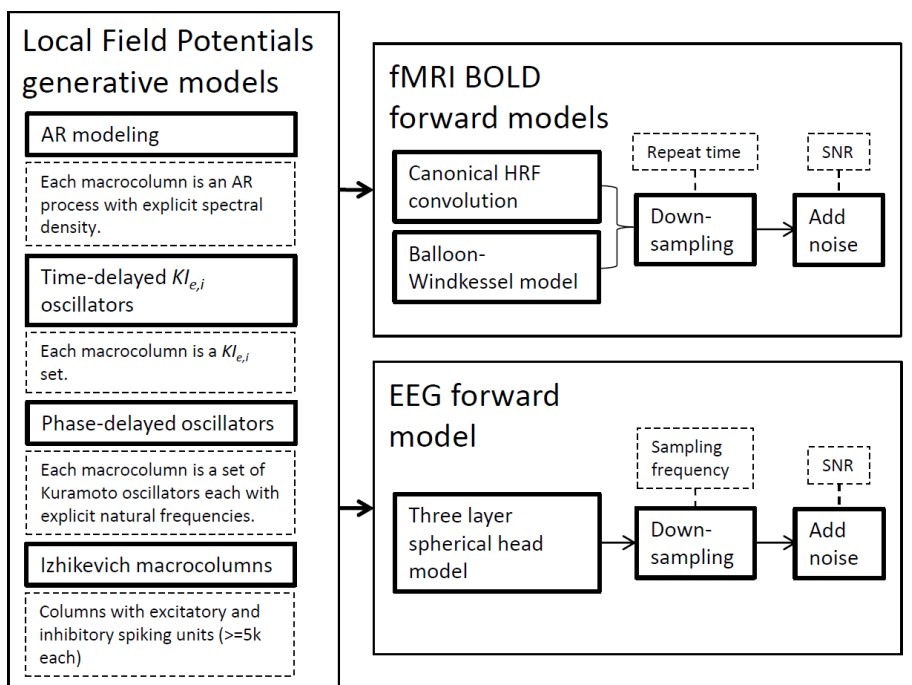

**Figure 1 Strategies used for synthetic neural data modeling.** LFPs are simulated by four distinct generative models and the resulting time-series can be used by EEG or BOLD forward models to produce the respective signals.

Simulations ran on a desktop PC equipped with an Intel® Core™ i7-2600K CPU @3.4 GHz, 8 GB of RAM and an NVIDIA® GeForce® GTX 580 GPU with 3 GB graphics memory. Simulations of Izhikevich columns ran on modified CUDA/C + + routines from CARLsim 2.0 (*Richert et al., 2011*) and used the parallel processing capabilities of the GPU. The remaining generative and forward models were implemented and ran in Matlab® and used only the CPU without any explicit parallelization.

## Autoregressive modeling

Autoregressive (AR) modeling is the simplest and most straightforward method for simulating neuronal datasets. By specifying the MVAR equations and parameters it is possible to simulate datasets from networks with different number of nodes, topology, noise, interaction delays and duration. Although it has been shown that the AR model's impulse response function has a transfer function that resembles the transfer function of a simple physiological model of EEG generation (*Blinowska & Franaszczuk, 1989*), as most connectivity metrics are parametric, estimating an MVAR model for datasets generated by linear AR processes might not pose the required challenge expected when presenting a novel connectivity metric. Nevertheless, this method has been used in many well-known works either standalone or as initial validation, followed by real neurobiological or neuroimaging datasets. It has been used to demonstrate that partial directed coherence (PDC) can correctly identify the directed connectivity in multivariate datasets from elaborate networks with reciprocal connections and interaction delays with different

magnitudes and durations (*Baccalá & Sameshima, 2001*). Similarly, the direct directed transfer function specific sensitivity to directed causal effects was also demonstrated with data from a MVAR model with unitary delays between variables (*Korzeniewska et al., 2003*). Equivalent models have been used to further study metrics like directed transfer function (DTF) (*Eichler, 2006*), improvements on spectral GC (*Chen, Bressler & Ding, 2006*) or demonstration of connectivity metrics in a vastly used toolbox (*Seth, 2010*).

MVAR models have also been key in data simulation for the still open debate (*Smith et al., 2010*; *Valdes-Sosa et al., 2011*; *Seth, Chorley & Barnett, 2013*) concerning the use of directed connectivity metrics to assess causal influences from fMRI BOLD signals. Roebroeck used synthetic data generated from a bi-dimensional first-order AR process as LFPs which were later transformed into BOLD signals in which causality was assessed with GC for different experimental parameters (*Roebroeck, Formisano & Goebel, 2005*). Later, Schippers used the same MVAR model to understand the effect of hemodynamic lag opposing the interaction neuronal delay in group analysis (*Schippers, Renken & Keysers, 2011*) and Barnett used a similar first order model to solve GC analytically after digital filtering (*Barnett & Seth, 2011*). Other studies used MVAR models to study the effectiveness of GC applied to cluster sets obtained with independent component analysis (*Londei et al., 2006*) and principal component analysis with partial canonical correlation analysis (*Sato et al., 2010*). Others have used AR networks where real LFP propagate according to predefined weights and delays to study the dependence of GC metrics to experimental parameters (*Deshpande, Sathian & Hu, 2009*; *Rodrigues & Andrade, 2014*).

An MVAR model of order $p$ can be defined by $X(t) = \sum_{j=1}^{p} \Phi_j X(t - j - D) + w(t)$ where $X(t)$ is the multivariate time-series, $\Phi_j$ is the MVAR coefficient matrix for time-lag $j$, $p$ is the model order, $D$ is the interaction delay matrix and $w(t)$ is the innovation matrix consisting in independent white noise. With these parameters, it is possible to define all the aforementioned network properties.

Here we also consider the possibility of defining the spectral peak frequency of the AR process that generates the activity of each variable. Considering the AR model for variable $n$, without accounting for inter-variable interactions, $X_n(t) = \sum_{j=1}^{p} \varphi_{nj} X_n(t - j) + w_n(t)$ a spectral peak frequency can be identified if the model is causal which requires all the roots of the AR polynomial $\varphi_{nj}(z) = 1 - \sum_{j=1}^{p} \varphi_{nj} z^j$ to be outside the unit circle. As proved by *Jiru (2008)*, when the absolute value of the AR coefficient for the time-lag $p$ is close to unity, $|\varphi_p| \approx 1$, the spectral peak frequencies have approximately the same values as the arguments of the roots of the AR polynomial. Other properties, especially for models with $p = 2, 3$ can be found in *Jiru (2008)*. The possibility to manipulate the spectral peak frequencies of each variable in the MVAR model adds up to the previous parameters so now it is possible to generate synthetic data for a given number of variables with a known connectivity pattern (reflected in the inter-variable AR parameters distribution), interaction delays (reflected in $D$), interaction strength (reflected in the inter-variable AR parameters value), SNR (reflected in $\sigma_w^2$) and peak frequency (reflected in the intra-variable parameters).

Although the inter-variable AR coefficients define the interaction strength between variable, it is also possible to compute the required values for these parameters in order to obtain a desired theoretical GC as done in *Barnett & Seth (2011)*. This is done for each variable pair by analytically deriving the expression for their theoretical GC value, based in the MVAR coefficient matrix, transfer function and noise covariance matrix, and solving it for the inter-variable coefficients.

For our simulations, a bivariate AR(2) model was created with variable 1 exerting causal effect in variable 2 at a specific frequency $\omega$, with variable intensity $F_{X_1 \rightarrow X_2}$ and delay $d_{21}$ following the expression:

$$X_1(t) = \varphi_{11}(2)X_1(t-2) + \varphi_{11}(1)X_1(t-1) + w_1(t)$$
$$X_2(t) = \varphi_{22}(2)X_2(t-2) + \varphi_{22}(1)X_2(t-1) + \varphi_{21}(1)X_1(t-d_{21}) + w_2(t). \tag{1}$$

Here $w_1$ and $w_2$ are the model's uncorrelated, zero mean, unit variance, white Gaussian innovation processes. Causality was chosen to occur always in the gamma band, more specifically at 33 Hz so, the AR parameters for variable 1 AR process were chosen so the spectral peak occurs at this frequency. Following the relationship $\omega = \pm \arccos(\varphi_{11}(1)(\varphi_{11}(2) - 1)/(4\varphi_{11}(2)))$ between spectral peak frequency and AR parameters found in *Jiru (2008)* for AR(2) models, variable 1 parameters must be $\varphi_{11}(1) = 1.337$ and $\varphi_{11}(2) = -0.98$ for a spectral peak to exist at 33 Hz.

The causal effect occurs due to $\varphi_{21}(1)$. To establish a relationship between this parameter and the consequent GC value, spectral GC can be solved analytically for the model (1) similarly to what occurs in *Barnett & Seth (2011)*. Skipping intermediate steps this results in:

$$F_{X_1 \rightarrow X_2}(\omega)$$
$$= \ln\left(1 + \frac{\varphi_{21}(2)^2}{\varphi_{11}(1)^2 + \varphi_{11}(2)^2 + 2\varphi_{11}(1)\cos(\omega)(\varphi_{11}(2) - 1) - 2\varphi_{11}(2)\cos(2\omega) + 1}\right). \tag{2}$$

Solving (2) for $\varphi_{21}(1)$ allows an AR(2) model to be built with the desired causality. Although LFPs simulation might seem redundant (GC estimation from data modeled with parameters obtained by the analytical solution of the spectral GC formulation) this data is helpful to analyze the effects of the forward EEG and BOLD models and to benchmark or demonstrate other causality metrics. Varying theoretical causality is modeled ($F_{X_1 \rightarrow X_2}(33 \text{ Hz}) = $ [from 0 to 5 in steps of 0.5], $d_{21} = 20$ ms) at 33 Hz for different interaction delays ($F_{X_1 \rightarrow X_2}(33 \text{ Hz}) = 5$, $d_{21} = $ [4 ms, and 20 to 100 ms in steps of 20 ms]). Initial values were randomly assigned, the simulations produced 60 s of data and the first 20 s were discarded to remove transient effects, data was generated at $1k$ Hz and subsampled to 250 Hz. Each 60 s of data required ~1 s of simulation time.

## Coupled oscillators

Coupled oscillator networks are a more realistic way to represent the dynamics between neuronal populations. Each node usually represents a population of excitatory and inhibitory spiking neurons that exhibit oscillations with varying levels of synchrony in

specific frequency ranges. The network can comply with one of two following dynamical regimes: a synchronous state with self-sustained oscillations (*Börgers & Kopell, 2003*) or in an asynchronous state with transient oscillations (*Mattia & Del Giudice, 2002*). These networks have been used to show that DTF can be interpreted within the GC framework (*Kamiński et al., 2001*) by modeling interacting cortical columns with excitatory and inhibitory populations with delay-coupled nonlinear stochastic differential equations (SDE). Similar SDE delayed networks, called Wilson–Cohen models, are also used in *Deco, Jirsa & McIntosh (2011)* to study the dynamics of simulated resting state networks (RSNs). A similar but more detailed dynamic causal model is used to study GC (*Friston et al., 2014*) where each cortical column is comprised of pyramidal and inhibitory cells from supra-granular layers, excitatory spiny cells in granular layers and deep pyramidal cells in infra-granular layers.

A general network of coupled second order differential equations similar to *Kamiński et al. (2001)*, represents a node as a system with delay-coupled nonlinear SDE for the excitatory population defined by:

$$\ddot{x}_n + (a+b)\dot{x}_n + abx_n(t) = -k_{ei}S(y_n(t)) + \sum_{m=1}^{N} k_{nm}S(x_m(t - d_{nm})) + I_n(t) + w_{x_n}(t) \qquad (3)$$

and inhibitory population defined by:

$$\ddot{y}_n + (a+b)\dot{y}_n + aby_n(t) = k_{ie}S(x_n(t)) + w_{y_n}(t). \qquad (4)$$

Variables $x_n$ and $y_n$ represent the LFPs of excitatory and inhibitory populations respectively of node $n$, $k_{ei}$ and $k_{ie}$ their respective coupling coefficients, $k_{nm}$ is the coupling coefficient from node $n$ to node $m$, $d_{nm}$ is the delay from node $m$ to node $n$, $a$ and $b$ are time constants that define the rate at which activity decays without input and, $w_{x_n}$ and $w_{y_n}$ are the independent white noise processes for nodes $n$ and $m$ respectively. $I_n(t)$ is the external input to the excitatory population and $S$ is the following sigmoidal function for a modulatory parameter $S_m$:

$$S(x, S_m) = \begin{cases} S_m(1 - e^{-(e^x - 1)/S_m}) & \text{if } x > -u_0 \\ -1 & \text{if } x \le -u_0 \end{cases} \qquad (5)$$
$$u_0 = -\ln(1 + \ln(1 + 1/S_m)).$$

Following the notations in *Freeman (1987)*, (3) and (4) is a coupling between $KO_i$ and $KO_e$ subsets; hence, each node can be seen as a $KI_{e,i}$ set. In this work, these parameters were used with the same values as in *Freeman (1987)* and *Kamiński et al. (2001)*: $a = 0.22/\text{ms}$, $b = 0.72/\text{ms}$, $k_{ei} = 0.4$, $k_{ie} = 0.1$, $S_m = 5$ and the independent Gaussian white noise processes had zero mean and 0.04 variance. The system's numerical solution was approximated with a fourth order Runge–Kutta method (delays where linearly interpolated for the intermediate increments) using a time-step of 0.1 ms, with the noise term being integrated with the Euler method using the same time-steps. The LFPs were initialized as zero for as long as the longest delay present in the simulation required.

A network of two $KI_{e,i}$ sets, defined by the interactions in (3) and (4), was simulated for 60 s with varying values of weak and strong coupling ($k_{21} = [0, 0.1, 0.2, 0.5, 0.7, 1, 3, 5, 10, 15, 22, 30]$, $d_{21} = 20$ ms) and interaction delay ($k_{21} = 30$, $d_{21} = [4$ ms and 20–100 ms in steps of 20 ms]) between sets 1 and 2. The first 20 s were discarded to remove transient effects, and data was generated at $10k$ Hz and subsampled to 250 Hz. External input $I_n(t)$ lasted 1 ms and had 1% probability of occurrence for each set. Each 60 s of data required ∼181 s of simulation time.

As *Friston et al. (2014)* concludes that GC is not appropriate for data generated by delay-coupled oscillators with unstable modes, and because self-sustained oscillations occur in large scale simulations (*Deco, Jirsa & McIntosh, 2011*), we also focus our simulations on networks functioning in the synchronous mode. In this regime, since oscillators show a limit cycle phase space trajectory, this phase can be modeled by a single dynamical variable reducing the former models to a simpler phase oscillators where it is possible to define the oscillating frequency. *Cabral et al. (2011)* modeled RSNs with several delayed-phase oscillator networks using the Kuramoto model (*Kuramoto, 1984*), layered for each frequency of interest. Therefore, in this work, we use the same network building scheme consisting of several stacked layers of two coupled oscillators (one layer for each natural frequency) where each phase variable $\theta_n$ is governed by the following dynamical equation:

$$\dot{\theta}_n = \omega_n + k \sum_{m=1}^{N} C_{nm} \sin(\theta_m(t - d_{nm}) - \theta_n(t)) + w_{\theta_n}(t). \tag{6}$$

Here $\omega_n$ is the angular frequency of each oscillator (rad/s), $C_{nm}$ is the relative coupling coefficient from node $m$ to node $n$, $k$ is a global coupling coefficient and the remaining variables represent the same parameters as in the previous dynamic equations. The neuronal activity can be obtained as the firing rate of the population of neurons represented by each oscillator. Following the procedure in *Cabral et al. (2011)*, the firing rate is the sine function $r_n = \sin(\theta_n(t))$ of phase variable.

For our simulations, the system's numerical solution was approximated with the Euler–Maruyama (*Kloeden & Platen, 1992*) method using a time-step of 0.1 ms. Phases were initialized randomly in the initial instants corresponding to the longest delay. A network of two nodes was simulated for 60 s (first 20 s were discarded to remove transient effects) by having three independent layers of pairs of phase-delayed coupled Kuramoto oscillators where at each layer share the same natural frequency and follow the dynamics in (6). Layers 1, 2 and 3 oscillate at 5, 33 and 60 Hz, respectively, and only layer 2 has a coupling coefficient between oscillators different than 0. Therefore, phase-delayed coupling can only occur at the 33 Hz and from node 1–2. Coupling varied from weak to strong ($k = [0, 0.1, 0.2, 0.5, 0.7, 1, 3, 5, 10, 15, 22, 30]$, $d_{21} = 20$ ms) and interaction delays followed the same values as the previous simulations ($k = 30$, $d_{21} = [4$ ms and 20–100 ms in steps of 20 ms]). Each 60 s of data required ∼55 s of simulation time.

## Izhikevich columns

The former oscillator models can be simulated in greater detail by modeling their constituting spiking units individually. In the context of testing directed connectivity, this modeling was used in *Seth, Chorley & Barnett (2013)* to understand how GC's sensitivity is affected by vascular latencies opposing the neuronal lag in BOLD signal time-series sampled with decreasing time repeat (TR). Besides spiking units, their synapses were also modeled with explicit NMDA, AMPA, GABAa and GABAb conductances and short-term plasticity (STP). The Izhikevich spiking model is able to produce several firing patterns observed in real neurons without the computational demand of more biophysically realistic models like the Hodgkin–Huxley, which makes it appropriate for larger scale simulations (*Izhikevich, 2003*). This model replaces the bio-physiologic meaning of the Hodgkin–Huxley model's equations and parameters by a topologically equivalent phase dynamic modeled with two ordinary differential equations and four parameters:

$$\dot{v} = 0.04v^2 + 5v + 140 - u + I$$
$$\dot{u} = a(bv - u). \tag{7}$$

Also, the additional after-spike resetting when $v \geq +30$ mV, $v = c$ and $u = u + d$.

The variable $v$ represents the neuron's membrane potential and $u$ its recovery variable. Parameters $a$, $b$, $c$ and $d$ are defined in order to implement one of the twenty spiking neurons shown in *Izhikevich (2004)*. Similarly to *Seth, Chorley & Barnett (2013)*, in this work we simulate cortical columns with two different neurons: regular spiking excitatory pyramidal neurons ($a = 0.02$, $b = 0.2$, $c = -65$, $d = 8$) and fast spiking inhibitory interneurons ($a = 0.1$, $b = 0.2$, $c = -65$, $d = 2$). Due to the model's simplicity, it is also computationally amenable to compute the synaptic input to each neuron with the neuro-transmitter conductances for each receptor type ($g_{AMPA}$, $g_{NMDA}$, $g_{GABAa}$ and $g_{GABAb}$):

$$I = g_{AMPA}(v - 0) + g_{NMDA}\frac{[(v+80)/60]^2}{1 + [(v+80)/60]^2}(v - 0)$$
$$+ g_{GABAa}(v - 70) + g_{GABAb}(v - 90). \tag{8}$$

These conductances are modeled with spike timing dependent plasticity (STDP) by being affected by incoming spike's origin and timing. Therefore, spikes incoming from excitatory neurons can change $g_{AMPA}$ and $g_{NMDA}$, while spikes incoming from inhibitory neurons can change $g_{GABAa}$ and $g_{GABAb}$. This update depends upon the difference between the timing of the post and pre-synaptic spikes, $\Delta t$, and two time constants for slow ($\tau = 0.1$ ms) and fast ($\tau = 0.01$ ms) synapses and is modeled exponentially: increment $\propto e^{-\Delta t/\tau}$ if $\Delta t > 0$ and decrement $\propto -e^{\Delta t/\tau}$ if $\Delta t \leq 0$.

STP is also modeled in the synaptic weights, influenced by pre-synaptic activity, with the scale factor $s(t)$:

$$s(t) = x(t)u(t)$$
$$\dot{x} = \frac{1 - x(t)}{t_D} - u(t)x(t)\delta(t - t_{\text{spike}})$$
$$\dot{u} = \frac{U - u(t)}{t_F} - U[1 - u(t)]\delta(t - t_{\text{spike}}). \tag{9}$$

Here $\delta$ is the Dirac function, and the state variables $x(t)$ and $u(t)$ have baseline levels of 1 and $U$, respectively. Parameters $t_D$ and $t_F$ are the depression and facilitating times which govern how fast $x(t)$ and $u(t)$ return to baseline. Excitatory synapses' STP is modeled with $U = 0.5$, $t_D = 800$ and $t_F = 1,000$ while inhibitory synapses have $U = 0.2$, $t_D = 700$ and $t_F = 20$. A small subgroup of excitatory neurons is stimulated by pre-synaptic spikes distributed randomly in time without STP or STDP.

At each column, LPF is obtained assuming dendritic AMPA currents as a good indicator of this activity by the average AMPA conductance in all afferent excitatory synapses (*Seth, Chorley & Barnett, 2013*).

In this work, we simulated two columns of $5k$ randomly distributed Izhikevich neurons (80% excitatory, 20% inhibitory) with STDP and STP between neurons of the same column and without both learning strategies in inter-column connections. Five percent of these excitatory neurons are stimulated by pre-synaptic spikes distributed randomly in time without STP or STDP. Instead of a single parameter to model the synaptic strength between columns, like the coupling coefficients or the inter-variable AR coefficient seen in the previous models, here it is possible to model three parameters related to inter-column connection strength. These are the percentage of neurons in one column that project to another column $p_{\text{projection}}$, the percentage of connections each branch does in the target column $p_{\text{branch}}$ and the synaptic strength of the individual connections $k_{\text{unit}}$. Simulations produced 60 s of data (the first 20 s were discarded to remove transient effects) with causal influence from column 1 to column 2. In this case, as three parameters define the coupling strength, the simulations were done for each one individually: for varying $p_{\text{projection}}$ ($p_{\text{projection}} = [0 \text{ to } 1 \text{ in steps of } 0.2]$, $p_{\text{branch}} = 0.05$, $k_{\text{unit}} = 1$, $d_{21} = 20$ ms), for varying $p_{\text{branch}}$ ($p_{\text{projection}} = 0.2$, $p_{\text{branch}} = [0.01, 0.05, 0.1, 0.3, 0.6, 1]$, $k_{\text{unit}} = 1$, $d_{21} = 20$ ms) and for varying $k_{\text{unit}}$ ($p_{\text{projection}} = 0.2$, $p_{\text{branch}} = 0.05$, $k_{\text{unit}} = [0.01, 0.05, 0.1, 0.3, 0.6, 1]$, $d_{21} = 20$ ms). In conformity with the previous generative models, simulations for different interaction delays were also performed $d_{21}$ ($p_{\text{projection}} = 0.2$, $p_{\text{branch}} = 0.05$, $k_{\text{unit}} = 1$, $d_{21} = [4 \text{ ms and } 20\text{–}100 \text{ ms in steps of } 20 \text{ ms}]$ ). The spiking units are ate rest in their initial state. Each 60 s of data required $\sim$52 s of simulation time.

## EEG forward modeling

EEG is simulated with a simple forward model that relates the LFP as the activity of a current dipole to the surface potential measured at the scalp. This relation is obtained by representing the head as a multilayer surface, with each layer having constant isotropic conductivity. Although there are realistically shaped multilayer head models like the

**Peer**J

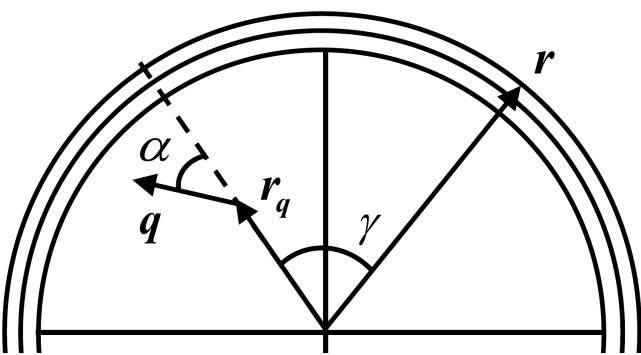

**Figure 2 Three layer spherical head model.** For one current dipole with radius $\mathbf{r_q}$ and moment $\mathbf{q}$ and scalp electrode with radius $\mathbf{r}$ oriented with angles $\alpha$ and $\gamma$, respectively. Adapted from *Mosher, Leahy & Lewis (1999)*.

finite-element method or the boundary element method (*Mosher, Leahy & Lewis, 1999*; *Fuchs et al., 2002*; *Darvas et al., 2004*), these are more important in the EEG inverse problem solution as they mitigate the distortion produced by simpler models (*Ermer & Mosher, 2001*). As the scope of this work is to simulate a generic EEG signal, we can adopt a simpler model of the skull like the three layer sphere with isotropic conductivities (*Berg & Scherg, 1994*) where the problem of volume conduction is observed. Figure 2 depicts a typical three layer model with an electrode placed in the scalp with radius $\mathbf{r}$, an intracranial current dipole with radius $\mathbf{r_q}$ and moment $\mathbf{q}$ and respective angles. This experiment uses a setup with two dipoles placed beneath the three layers with 8 cm outer radius ($\sigma_{\text{brain}} = 0.33$ S/m $r_{\text{brain}} = 7.04$ cm, $\sigma_{\text{skull}} = 0.0042$ S/m, $r_{\text{skull}} = 7.44$ cm, $\sigma_{\text{scalp}} = 0.33$ S/m, $r_{\text{scalp}} = 8$ cm) spaced by 2 cm and two electrodes are placed in the scalp also spaced by 2 cm.

The solution $v^1(\mathbf{r}; \mathbf{r_q}; \mathbf{q})$ at radius $\mathbf{r}$ for the simplest case of a single spherical layer head model for a current dipole with moment $\mathbf{q}$ at radius location $\mathbf{r_q}$ can be obtained by the sum of the radial $v_r^1(\mathbf{r}; \mathbf{r_q}; \mathbf{q})$ and tangential $v_t^1(\mathbf{r}; \mathbf{r_q}; \mathbf{q})$ potentials:

$$
\begin{aligned}
v_r^1(\mathbf{r}; \mathbf{r_q}; \mathbf{q}) &= \left(\frac{q\cos\alpha}{4\pi\sigma}\right)\left(\frac{2(r\cos\gamma - r_q)}{d^3} + \frac{1}{r_q d} - \frac{1}{r r_q}\right) \\
v_t^1(\mathbf{r}; \mathbf{r_q}; \mathbf{q}) &= \left(\frac{q\sin\alpha}{4\pi\sigma}\right)\cos\beta\sin\gamma\left(\frac{2r}{d^3} + \frac{d+r}{rd(r - r_q\cos\gamma + d)}\right).
\end{aligned}
\tag{10}
$$

Here $\alpha$ is the angle the dipole expresses in relation to its location vector $\mathbf{r_q}$, as can be seen in Fig. 2, $\sigma$ is the conductivity of the shell, $d$ is the value of the direct distance between $\mathbf{r_q}$ and $\mathbf{r}$, and $\beta$ is the angle between the plane formed by $\mathbf{r_q}$ and $\mathbf{q}$, and the plane formed by $\mathbf{r_q}$ and $\mathbf{r}$.

Following (*Berg & Scherg, 1994*; *Zhang, 1995*) a three layer model can be approximated with good accuracy by single layer spheres with the approximation $v^3(\mathbf{r}; \mathbf{r_q}; \mathbf{q}) \cong v^1(\mathbf{r}; \mu_1\mathbf{r_q}; \lambda_1\mathbf{q}) + v^1(\mathbf{r}; \mu_2\mathbf{r_q}; \lambda_2\mathbf{q}) + v^1(\mathbf{r}; \mu_3\mathbf{r_q}; \lambda_3\mathbf{q})$. The $\mu$ and $\lambda$ are the "Berg parameters" and are used to create three new dipoles with locations consisting in scaling $\mathbf{r_q}$ by $\mu$ in its radial direction and scaling the moment $\mathbf{q}$ by $\lambda$. The approximation $v^3(\mathbf{r}; \mathbf{r_q}; \mathbf{q})$

is computed for each electrode and dipole present in the simulation. Noise is added to the resulting time-series.

Details about the computation of the "Berg parameters" can be found in *Zhang (1995)* and other methods for EEG forward models in *Mosher, Leahy & Lewis (1999)*, *Ermer & Mosher (2001)* and *Darvas et al. (2004)*. The Brainstorm application offers several methods for the EEG forward model among others (*Tadel et al., 2011*).

For our simulations, each generated LFP was fed to an EEG forward model with two radial dipoles spaced by 2 cm placed beneath the three layers with 8 cm outer radius ($\sigma_{brain} = 0.33$ S/m $r_{brain} = 7.04$ cm, $\sigma_{skull} = 0.0042$ S/m, $r_{skull} = 7.44$ cm, $\sigma_{scalp} = 0.33$ S/m, $r_{scalp} = 8$ cm) and two recording electrodes placed in the scalp also spaced by 2 cm. White Gaussian noise was added in order for a linear SNR of 10.

## BOLD forward modeling

The BOLD signal time-series is the result of a series of neuronal and vascular events that produce a measurable change in the blood hemoglobin concentration. It is therefore an indirect and noisy observation of the neuronal activity as during neuronal activation local vessels are dilated to increase the blood flow and with it, oxygen and glucose delivery. The increased metabolism results in a localized increase in the conversion of oxygenated hemoglobin to deoxygenated hemoglobin and BOLD fMRI uses the latter as the contrast agent (*Ogawa & Lee, 1990*). This activity can peak four seconds after the neuronal event onset, although this value varies within and between subjects (*Handwerker, Ollinger & D'Esposito, 2004*). Hence, the location, dynamics and magnitude of the BOLD signal's activity are vastly influenced by the local vascular bed. This, combined with the fact that fMRI scanners sample an entire volume with TR in the time scale of a second, raised the question if directed functional connectivity, which aims at detecting temporal precedence between neuronal events in the order of tens to hundreds of milliseconds (*Ringo et al., 1994*; *Formisano et al., 2002*; *De Pasquale et al., 2010*), can offer accurate measures from BOLD signals. The simulation of BOLD signals was an important factor to answering this question by allowing experimental control over neuronal and hemodynamic parameters, and this has been achieved mainly by convolution with a canonical hemodynamic response function (HRF) or by dynamic modeling of the vascular activity with the extended Balloon-Windkessel (BW) model (*Friston et al., 2000*).

The first approach started being used with one gamma function for the HRF convolution kernel (*Goebel, 2003*; *Roebroeck, Formisano & Goebel, 2005*) with the purpose of investigating the effect of filtering, down-sampling and noise in GC estimation. Following simulation studies (*Deshpande, Sathian & Hu, 2009*; *Schippers, Renken & Keysers, 2011*; *Seth, Chorley & Barnett, 2013*; *Rodrigues & Andrade, 2014*) started using a dual-gamma function as used in SPM software (*Friston, Holmes & Ashburner, 1999*) with parameters as time to peak, time to undershoot, onset time, dispersion of response, dispersion of undershoot and their ratio following the distributions found in *Handwerker, Ollinger & D'Esposito (2004)* to study the effects of down-sampling, noise and HRF variability. Compared to the convolution approach, the BW model is more biophysically

interpretable and can also present nonlinear neuro-vascular couplings although when used in simulation studies there were no considerable differences in GC estimates between both (*Smith et al., 2010*; *Seth, Chorley & Barnett, 2013*).

In this work, both approaches were used with the default parameters offered by SPM12 in the functions *spm_hdm_priors* for the BW model and *spm_hrf* for the canonical HRF, with a 0.5 TR and a linear SNR of 10. However, in these simulations, new LFPs had to be created for all generative models with different interaction delays, duration and connectivity frequency. Interaction delays are increased to 200 ms, in order to counteract the reduction in sensitivity due to the low sampling period of typical fMRI TRs and low SNR (*Deshpande, Sathian & Hu, 2009*), and the LFP length had to be increased due to the low sampling rate; hence, 300 s were simulated. For the generative models where it is possible to control the frequency where the causal influence is exerted these were set, from 33 Hz in the previous analysis, to 0.1 Hz in the Kuramoto oscillators and <1 Hz for the MVAR modeling.

## Spectral Granger causality

We used the standard Geweke's spectral decomposition for GC (GGC) (*Geweke, 1982*) to infer causality between synthetic time-series. Unlike other spectral directed functional connectivity metrics such as PDC or DTF, GGC is not bounded between 0 and 1, which is useful in this study to see how the increase in the connection strength between variables effects the absolute value of causality across the different simulation methods. GGC decomposes the GC index (GCI) (*Granger, 1969*) into frequency components additively, meaning that the sum of all the frequency components from zero to the Nyquist frequency result in the GCI. For bivariate time-series, which is the case in this study, GGC can be computed from a MVAR model parameters by:

$$F_{j \to i}(f) = \ln \frac{S_{ii}(f)}{S_{ii}(f) - \left( \Sigma_{jj} - \frac{\Sigma_{ij}^2}{\Sigma_{ii}} \right)|H_{ij}(f)|^2} \quad j, i \in \{1, 2\}. \quad (11)$$

Here $\Sigma$ is the covariance matrix of the model's errors, $H$ is the transfer matrix, and $S$ is the spectral matrix. The MVAR model order can be estimated with the Bayesian information criterion (BIC) (*Schwarz, 1978*) or with the Akaike information criterion (AIC) (*Akaike, 1974*).

For multivariate time-series, refer to *Chen, Bressler & Ding (2006)* for more details.

## RESULTS

This section presents the results of applying (11) to bivariate time-series simulated with the generative and forward models introduced in Fig. 1 and in the previous sections with varying interaction strength and delay. This analysis aims at finding whether these changes in the generative modeling are captured by standard causality estimation by analyzing how GGC's difference of influence (DOI) ($F_{j \to i} - F_{i \to j}$ when $F_{j \to i}$ is the true causal direction) varies with the modeled interaction strength, and how GGC's DOI and estimated MVAR model order estimation change with increasing interaction delays. An

increase in empirical model order from BIC or AIC suggests that the interaction delays introduced in the generative model are detected in the resulting time-series.

## LFP

LFPs represent the local activity of neuronal populations without forward modeling hence, without the confounding effects expected from fMRI BOLD signals or EEG data and, the only randomness is due to the stochastic parameters from each model. This way, this analysis tests each generative model's capability to produce synthetic datasets with detectable causal effects.

### AR modeling

Results for GGC with varying modeled causality and interaction delays can be seen in Figs. 3A and 4A, respectively, and the model order estimated with BIC and AIC for the different interaction delays are shown in Fig. 5A.

The estimated GGC DOI present in Fig. 3A shows, as expected, causal influence at 33 Hz with the same absolute value from what was modeled. Changing the interaction delay from 4 to 100 ms doesn't affect the estimated causality, as can be seen in Fig. 4A, and both BIC and AIC model orders increase linearly with the increase in interaction delay (Fig. 5A). Also, these model orders correspond exactly to the neuronal delay; with a sampling rate of 250 Hz, the neuronal delays in $d_{21}$ correspond to [1, 5–25 in steps of 5] lagged observations which equal the model orders suggested by BIC and AIC (except for the neuronal delay of 4 ms).

### Time-delayed coupled KI$_{e,i}$ sets

Figures 3B and 4B respectively show the estimated GGC for varying coupling and interaction delay and Fig. 5B the respective model orders estimated with BIC and AIC.

Figure 3B shows that there are two frequency bands where connectivity is detected, around 20 Hz and around 60 Hz. These causal influences are more expressive for strong couplings (>1) and are inexistent at weak couplings below 0.5. When the interaction delay changes, connectivity also changes its frequency (Fig. 4B). This occurs as different delays change the oscillatory behavior of these neuronal populations, and also because the MVAR model order also changes which influences the AR spectrum. In Fig. 5B it is possible to see that both BIC and AIC are sensitive to the increase in the interaction delay and overestimate the true model order by approximately 2 lagged observations.

### Phase-delayed coupled Kuramoto oscillators

Figures 3C and 4C respectively show the estimated GGC for varying coupling and interaction delay and Fig. 5C the respective model orders estimated with BIC and AIC

GGC only detects causal relations with positive DOI at 33 Hz as can be seen in Fig. 3C. However, these only present expressive values on strong couplings (>3). Different interaction delays do not affect the GGC intensity or frequency distribution (Fig. 4C) although the model order suggested by BIC and AIC is not linearly related to the interaction delay (Fig. 5C) and is overestimated.

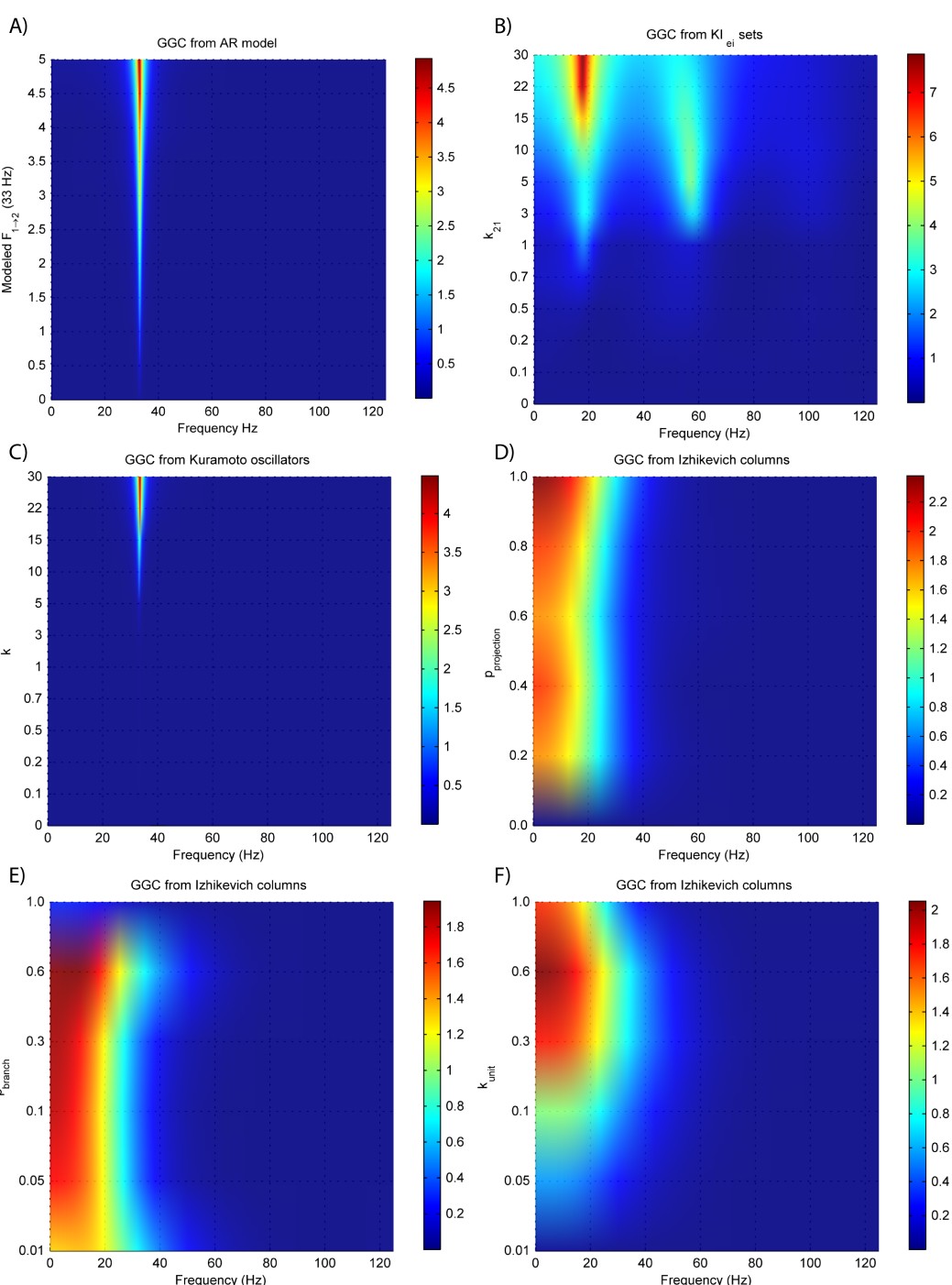

**Figure 3  GGC DOI for the time-series simulated with increasing coupling strength for the generative models.** (A) MVAR models, (B) $KI_{e,i}$ sets, (C) Kuramoto oscillators, (D)–(F) Izhikevich columns.

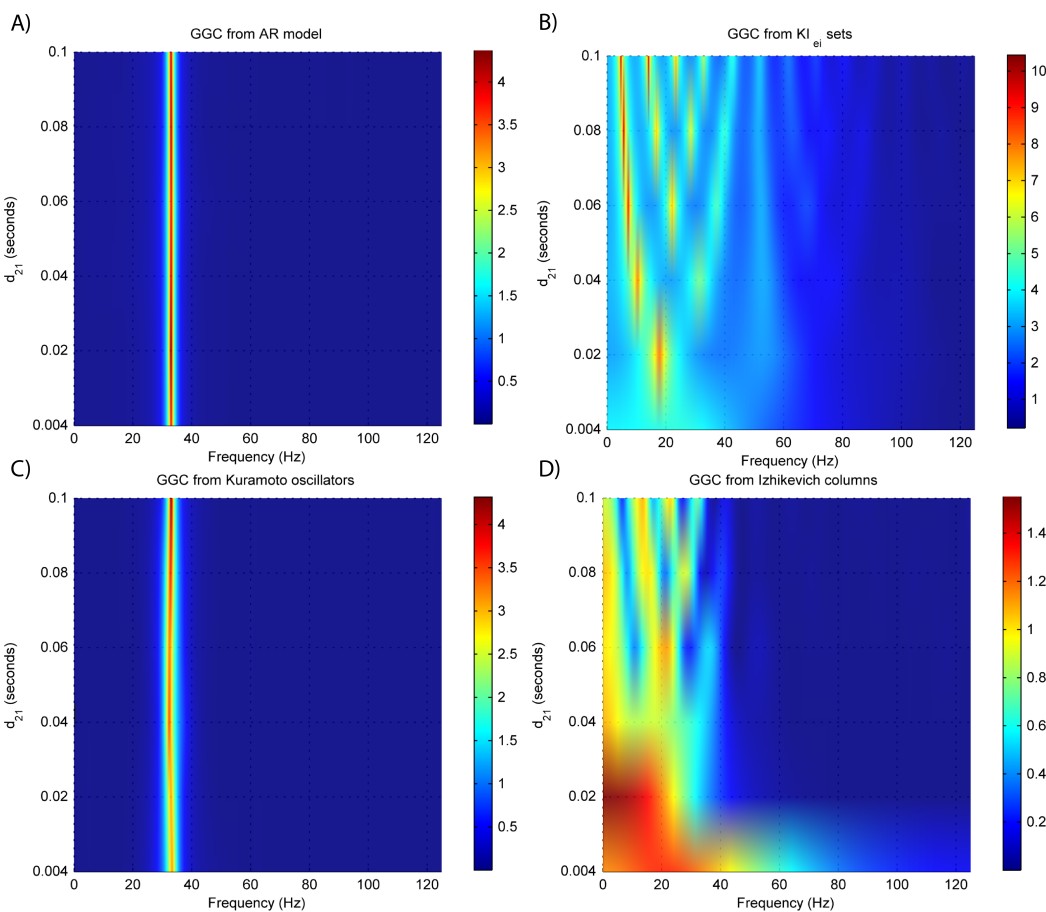

**Figure 4 GGC DOI for the time-series simulated with increasing interaction delay for the generative models.** (A) MVAR models, (B) $KI_{e,i}$ sets, (C) Kuramoto oscillators, (D) Izhikevich columns.

### *Izhikevich columns*

GGC results for varying $p_{\text{projection}}$ can be seen in Fig. 3D, results for varying $p_{\text{branch}}$ can be seen in Fig. 3E and results for varying $k_{\text{unit}}$ can be seen in Fig. 3F. GGC results for simulations with varying are shown in Fig. 4D and the respective model orders estimated with BIC and AIC in Fig. 5D.

From Figs. 3D–3F it is possible to see that GGC detects connectivity below 20 Hz and that a linear increase only occurs with $k_{\text{unit}}$. Interaction delays change the connectivity frequency distribution (Fig. 4D) as different delays change the oscillatory behavior of the neuronal populations and different MVAR model order produce different AR spectrums. Increasing the interaction delay produces a linear increase in model orders estimated with BIC although these are slightly overestimated (Fig. 5D).

## EEG forward modeling

Figure 6 shows the same experiments as Fig. 3 only this time, the time-series is the EEG recorded at the scalp electrodes. It is possible to see that, except for the Izhikevich columns, GGC amplitude is reduced.

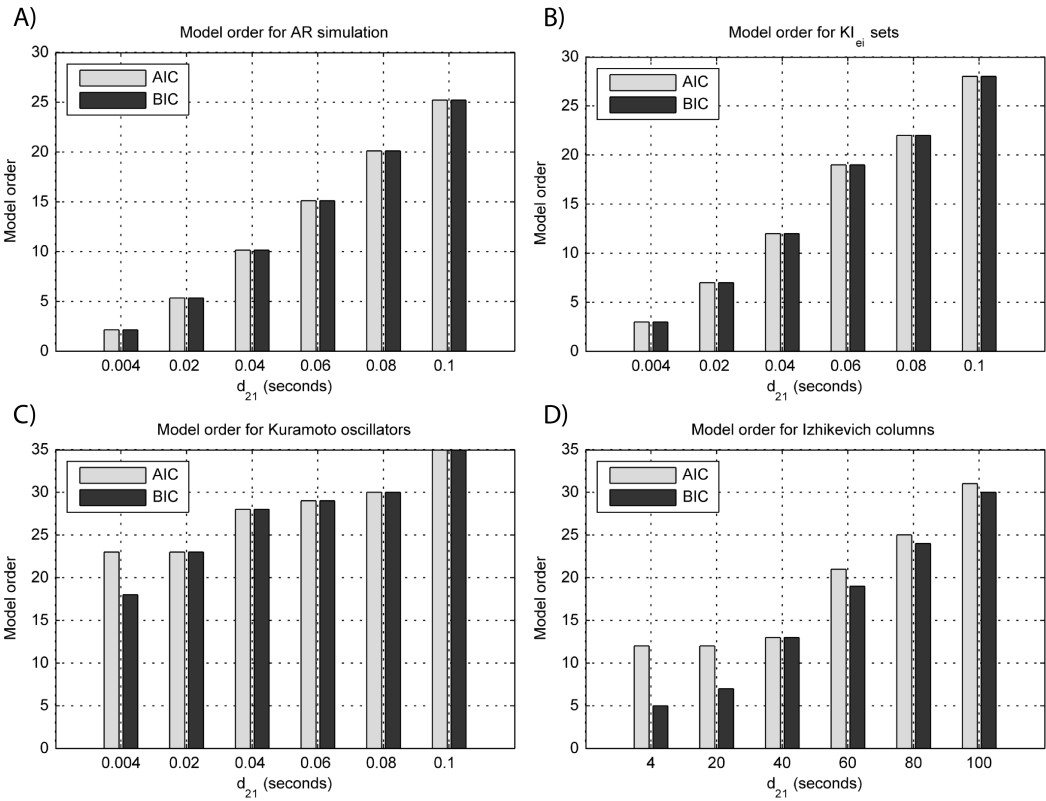

**Figure 5 Model orders estimated with BIC and AIC for the time-series simulated with increasing interaction delay for the generative models.** (A) MVAR models, (B) $KI_{e,i}$ sets, (C) Kuramoto oscillators, (D) Izhikevich columns. The lagged observations for these interaction delays with 250 HZ sampling rate are [1, 5, 10, 15, 20, 25].

EEG forward modeling does not affect the detection of interaction delays as can be seen in Fig. 7. However, for the $KI_{e,i}$ sets, causality around the 20 Hz dissipates and causality at 60 Hz is maintained. At some delays (0.4 and 0.8 ms) the EEG from the Kuramoto oscillators (Fig. 7C) shows GGC values similar to the LFPs.

Overall, model orders are overestimated for EEGs from every generative models. In the EEG from MVAR modeling and $KI_{e,i}$ sets (Figs. 8A and 8B, respectively) this is more pronounced in the lower interaction delays, and in the EEG from the Izhikevich columns (Fig. 8D) the opposite seems to occur. The overestimation problems in the LFPs from the Kuramoto oscillators are more pronounced after EEG forward modeling (Fig. 8C).

## BOLD forward modeling

The results for both the convolution with canonical HRF and the extended BW model are shown in Figs. 9 and 10.

The results in Fig. 9 show that GGC is greatly reduced after BOLD forward modeling regardless of the generative or forward models. In BOLD signals generated with MVAR modeling, causal effects are more visible after a modeled causality of 2.5 (Fig. 9A) although these remain around 0.5. The same values are achieved by $KI_{e,i}$ sets and Kuramoto

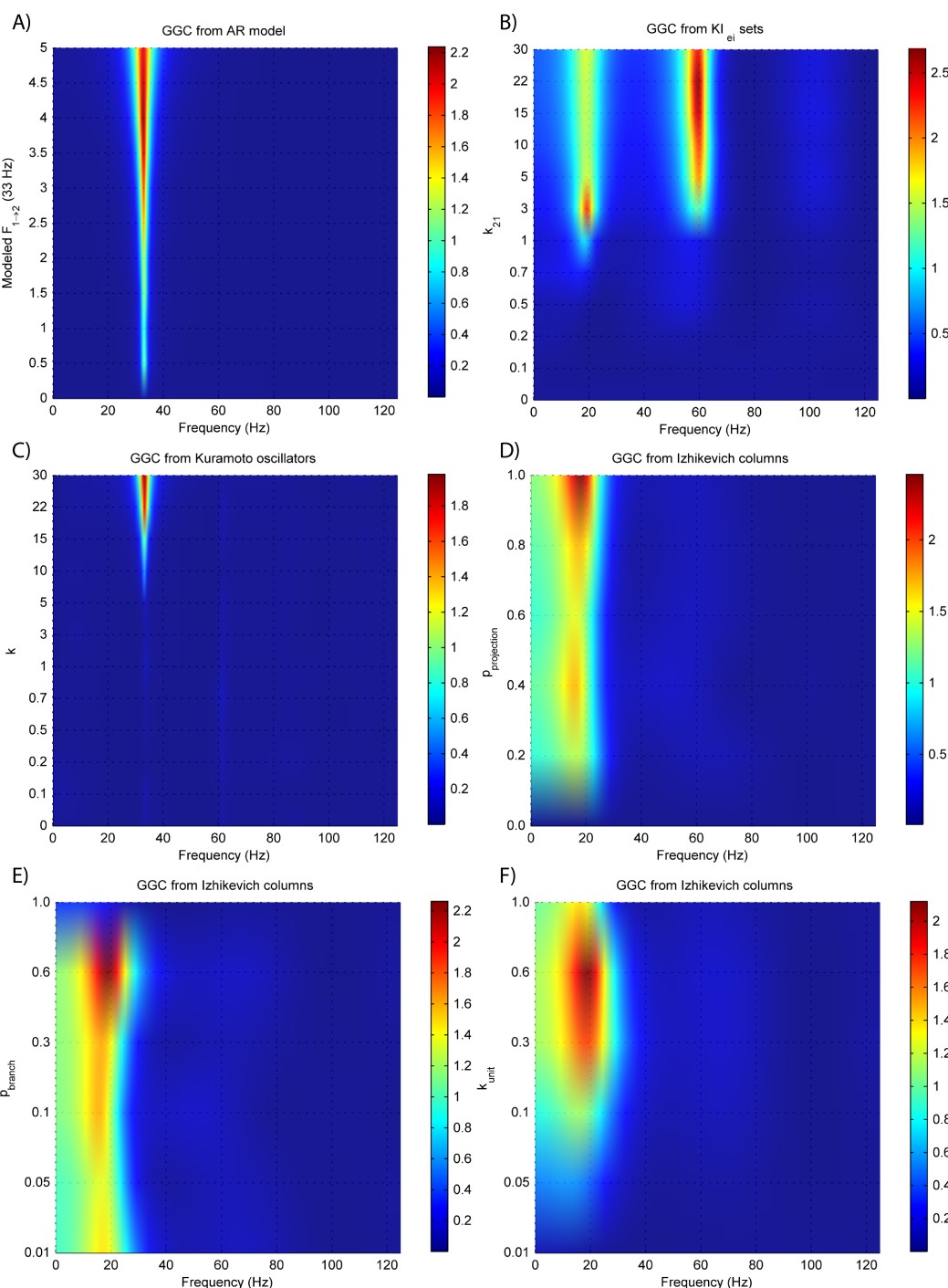

**Figure 6 GGC DOI after EEG forwarding the time-series from the generative models with varying coupling strengths.** (A) MVAR models, (B) $KI_{e,i}$ sets, (C) Kuramoto oscillators, (D)–(F) Izhikevich columns.
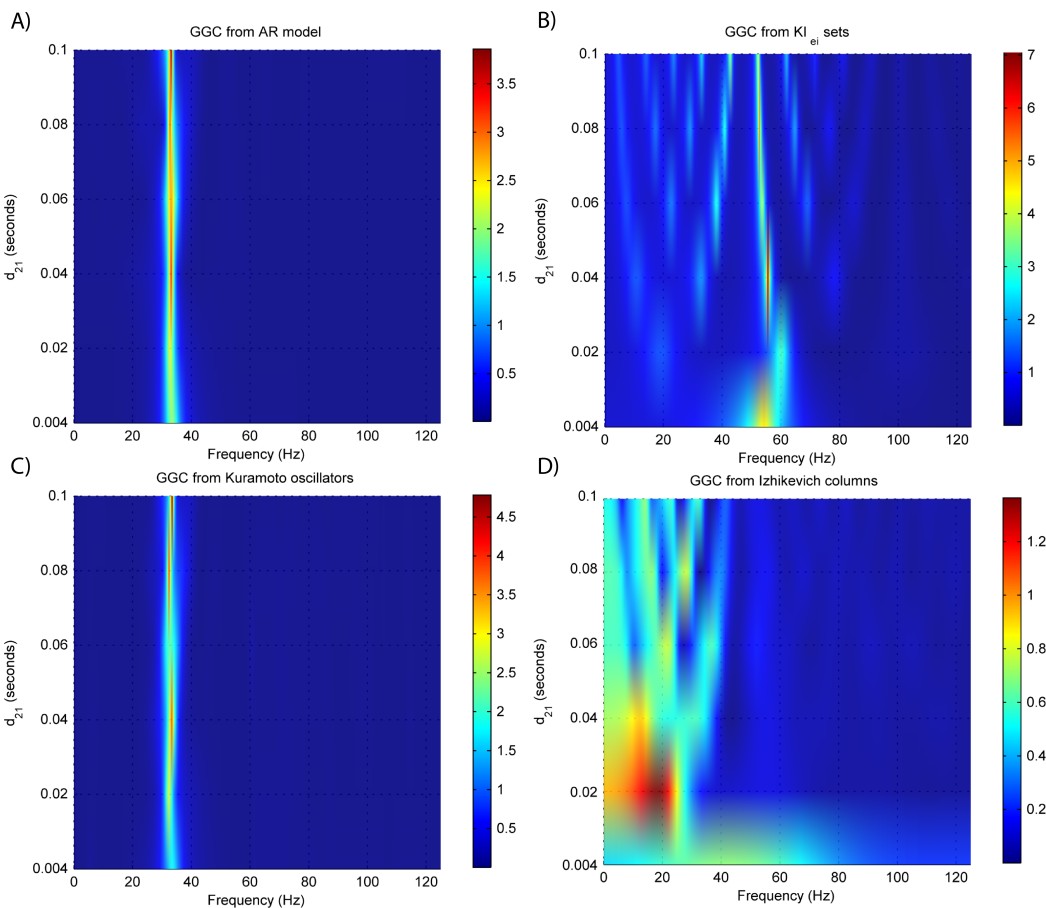

**Figure 7 GGC DOI after EEG forwarding the time-series from the generative models with varying interaction delays.** (A) MVAR models, (B) $KI_{e,i}$ sets, (C) Kuramoto oscillators, (D) Izhikevich columns.

oscillators after a coupling of 1 and 10 respectively. The Izhikevich columns only achieved these values for $p_{\text{projection}} > 0.4$, $p_{\text{branch}} > 0.6$ or $k_{\text{unit}} > 0.3$. Both the canonical HRF and the extended BW show similar results.

By keeping the coupling strengths at the maximum values in Fig. 9 and varying the interaction delay from 100 ms to 300 ms, the results in Fig. 10 suggest that all generative and forward models benefit from higher delays. With MVAR modeling, 150 ms is the value when causality is detectable (Fig. 10A) whereas for the remaining generative models this value is at 200 ms. Again, there are no relevant differences between canonical HRF and extended BW models.

## DISCUSSION

This study explores four generative models that represent distinct methodologies: multivariate MVAR modeling, neural mass models and spiking neuron populations. Although most of these models have been used previously in simulation studies that aim at benchmarking connectivity metrics, their capability to reflect causal interactions in their generated neuronal time-series had not been compared yet. Also, these neuronal

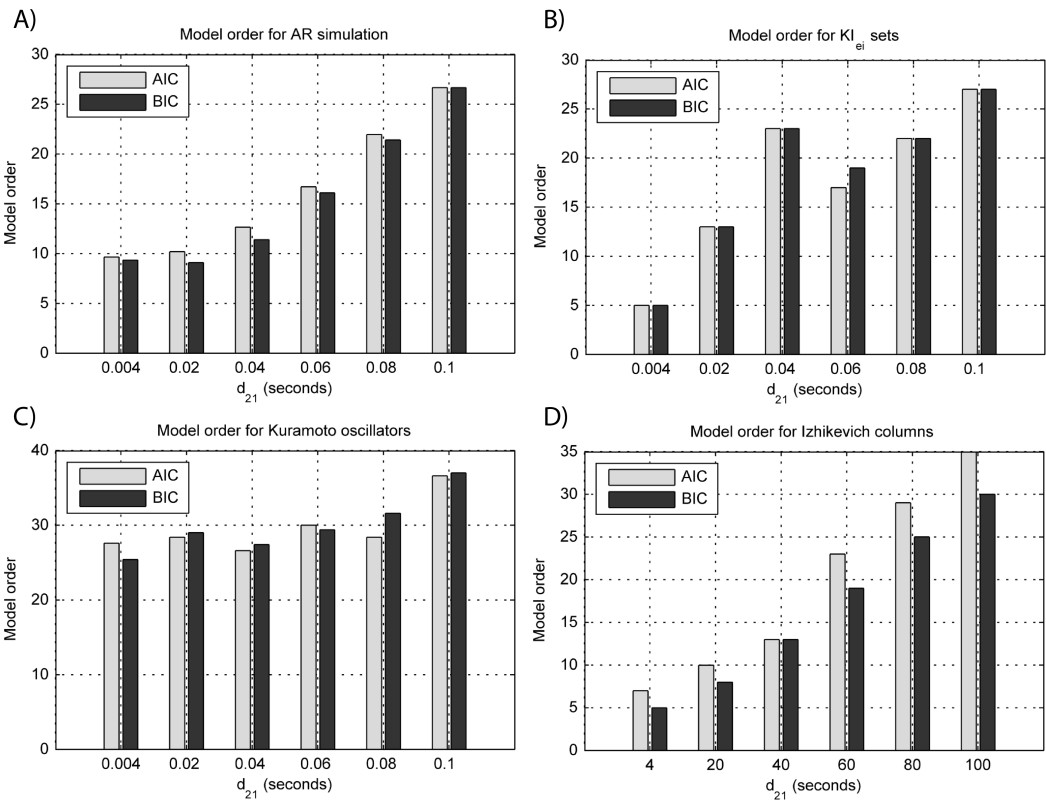

**Figure 8 Model orders estimated with BIC and AIC for the time-series simulated with increasing interaction delay after EEG forwarding with the generative models.** (A) AR models, (B) $KI_{e,i}$ sets, (C) Kuramoto oscillators, (D) Izhikevich columns. The lagged observations for these interaction delays with 250 HZ sampling rate are [1, 5, 10, 15, 20, 25].

time-series are not limited to the LFPs, BOLD signals and EEG covered by this work but these are the most widely used in connectivity studies.

Concerning the generative models in this work, both MVAR modeling and Kuramoto oscillators offer the possibility to directly specify the frequency of their oscillatory activity and therefore the frequency where connectivity occurs whereas in $KI_{e,i}$ sets and Izhikevich columns this is not possible, at least directly. On the other hand, $KI_{e,i}$ sets and Izhikevich columns are neurophysiologically plausible, as they offer the possibility to modulate different types of excitatory and inhibitory neurons. This suggests two different uses for these subgroups of generative models. MVAR modeling and Kuramoto oscillators, as these allow to manipulate the strength and frequency of causal relationships (or even the theoretical GGC in the MVAR modeling), seem more adequate for initial testing of directed functional connectivity metrics (*Baccalá & Sameshima, 2001*), studying the effect of post processing or other transformations in data prior to causality inference (*Barnett & Seth, 2011*) or to compare different metrics performance in an extended benchmark (*Rodrigues & Andrade, 2014*). Due to their superior neurophysiological realism, $KI_{e,i}$ sets and Izhikevich columns are more useful to inquire about the effectiveness of certain connectivity metrics for data recorded in specific brain locations with known dynamics

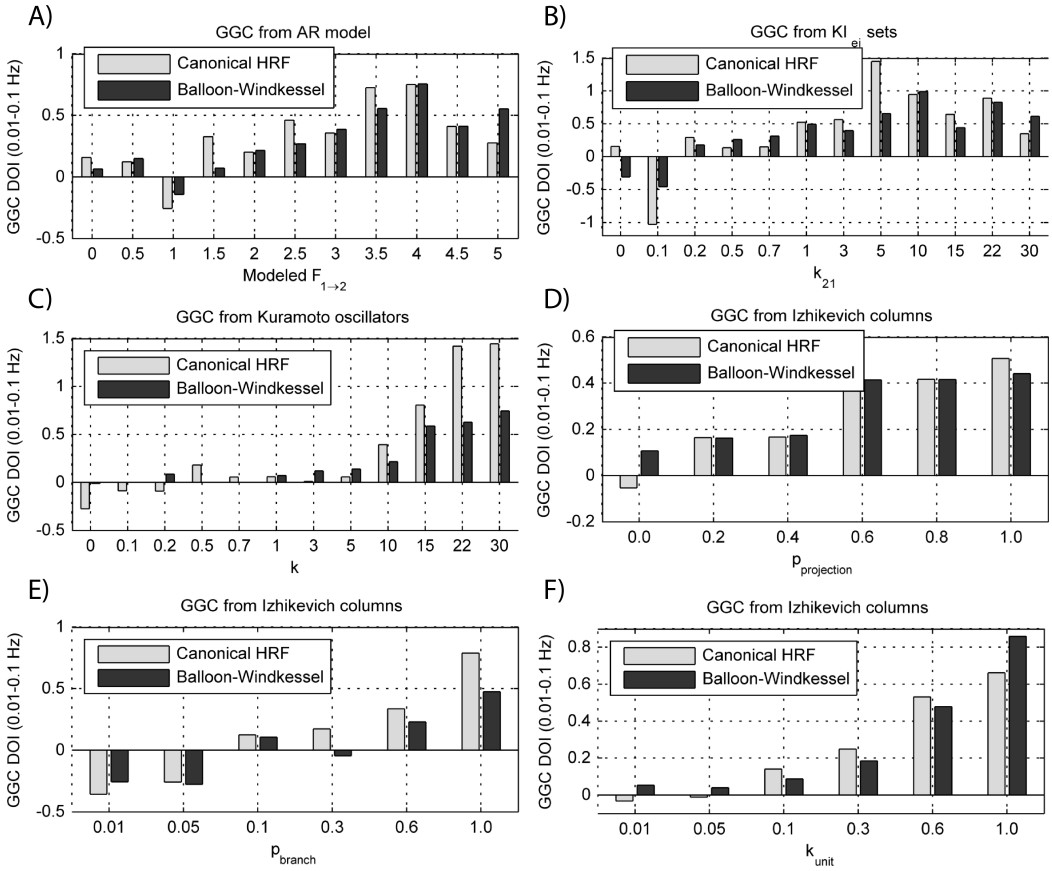

**Figure 9 GGC DOI in the 0.01–0.1 Hz band after BOLD forward modeling the time-series from the generative models with varying coupling strengths.** (A) MVAR models, (B) $KI_{e,i}$ sets, (C) Kuramoto oscillators, (D)–(F) Izhikevich columns.

(*Friston et al., 2014*). For example, in *Freeman (1987)* coupled *KI* sets are used to simulate chaotic EEG emanating from the olfactory system and in *Richert et al. (2011)* Izhikevich columns are used to simulate a large-scale model of cortical areas V1, V4, and middle temporal (MT) with color, orientation and motion selectivity.

All generative models are able to produce LFPs with detectable causal relations, and these results show the relation between experimental parameters and simulated causality. Only MVAR modeling allows for a direct specification of causality by solving the analytical equations for GGC applied to the MVAR coefficients. Concerning the neural mass models, the $KI_{e,i}$ sets show causal effects with lower coupling strength than the Kuramoto oscillators as the first start having causal relations with $k_{21} = 0.7$ while the second required values of $k > 3$ for identifiable causality. In the Izhikevich columns, the percentage of projecting neurons from the source column $p_{\text{projection}}$ is the variable with least influence in the observed GGC, except for when it is zero. Increases in both the percentage of target connections per projection $p_{\text{branch}}$ and the synaptic strength $k_{\text{unit}}$ lead to increases in the observed GGC. Although it wasn't tested in this work, it is possible that these values change for different number of neurons per column. In all generative

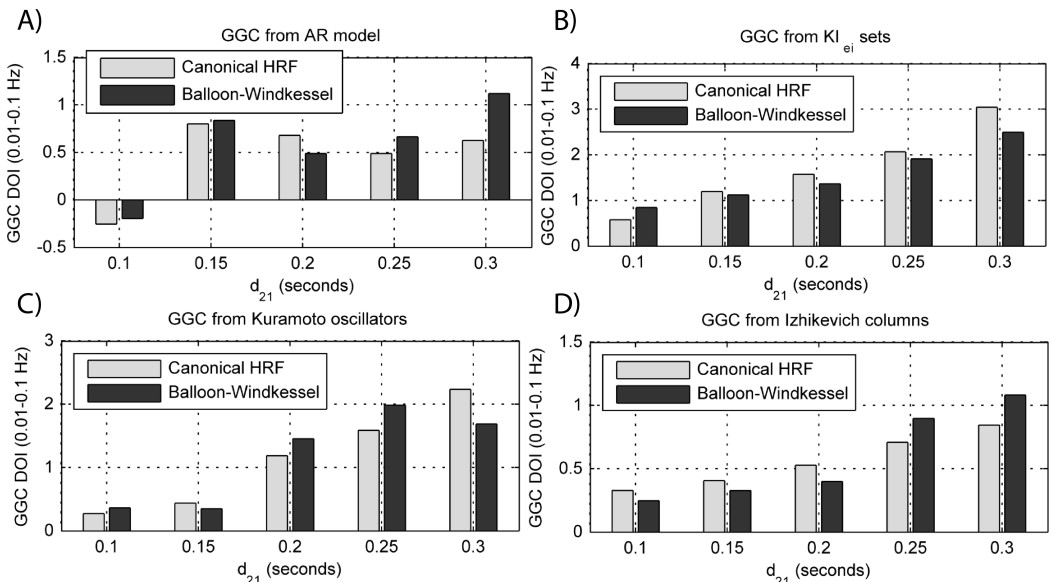

**Figure 10 GGC DOI in the 0.01–0.1 Hz band after BOLD forward modeling the time-series from the generative models with varying interaction delays.** (A) MVAR models, (B) $KI_{e,i}$ sets, (C) Kuramoto oscillators, (D) Izhikevich columns.

models the modulation of the interaction delay is possible without loss of causal relations, although the neurophysiologically realistic models show different frequency spectrums for different interaction delays. This neuronal delay is also detected by the BIC and AIC, which suggest higher model orders for higher interaction delays.

Forward models lead to a decrease in the absolute value of GGC, especially the forward BOLD model where negative DOIs could be found in worst scenarios. EEG forward modeling reduces the estimated GGC due to the added noise and to the volume conduction effect that produces a "cross-talk" between the two neural populations and electrodes; these effects would be more adverse if more neuronal populations were added inside the three-shell sphere. Nevertheless, these results show that all generative models produce LFPs that, when treated as activity from radial dipoles, preserve the causal relations in the resulting EEG. BOLD forward modeling is more detrimental to the preservation of causal effects, and this work did not model hemodynamic variability between brain regions (*Handwerker, Ollinger & D'Esposito, 2004*) which would further affect causality preservation in BOLD signal time-series (*Deshpande, Sathian & Hu, 2009*). Causality was reduced approximately by a factor of 10 in all generative models, and there were no concise differences between HRF convolution and BW modeling except for the Kuramoto models where the last leads to smaller values of causality.

The overview from Table 1 confirms our initial suggestions that AR models are suitable for exhaustive benchmarks of causal measures to study their dependence on experimental parameters and formulation, due to their low computational load associated with versatility and analytical solution. For benchmarks with fewer experimental parameters and increased concern in emulating neuronal data from frequency specific synchronized

**Table 1 Summary of the main results and characteristics for the four generative models.**

| | AR models | $KI_{e,i}$ sets | Kuramoto oscillators | Izhikevich columns |
|---|---|---|---|---|
| Causal relations with different strengths | yes | yes | yes | yes |
| Causal relations with different delays | yes | yes | yes | yes |
| Frequency specific causal relations | yes | no | yes | no |
| Analytic calculation of theoretical causality | yes | no | no | no |
| Neurophysiological model | no | yes | yes | yes |
| Can simulate different neuronal dynamics | no | no | yes | yes |
| Causality is preserved after EEG forwarding | yes | yes | yes | yes |
| Causality is preserved after hemodynamic forwarding | yes | yes | yes | yes |
| Expected computational load | very low | high | average | average |

populations, the Kuramoto oscillators offer the best compromise between versatility and computational load. Finally, to study causality in specific known neuronal dynamics both the $KI_{e,i}$ sets or the Izhikevich are the most appropriate thanks to their ability to realistically simulate varied neuronal populations.

## CONCLUSION

This work presented and analyzed different modeling strategies to generate artificial neuronal datasets for benchmarking purposes. LFPs are obtained by generative models and can be used by forward models to produce other recordings of neuronal activity such as the BOLD signal or EEG. All the analyzed models were able to transmit their causal structure ($\varphi_{21}(1), k_{21}, k, p_{\text{projection}}, p_{\text{branch}}, k_{\text{unit}}$) into their generated data but with different relations between these and the identified GGC. This study only covered bivariate models, but the same analysis could be performed with larger networks with large-scale fluctuations (*Cabral et al., 2011*). This would be useful to identify the directed functional connectivity metrics most appropriate to analyze large scale data such as fMRI BOLD signals from resting state networks.

### Funding

The Portuguese Foundation for Science and Technology (FCT) provided financial support through Project PTDC/SAU-ENB/112294/2009 and PEst-OE/SAU/UI0645/2014. The funders had no role in study design, data collection and analysis, decision to publish, or preparation of the manuscript.

### Grant Disclosures

The following grant information was disclosed by the authors:
Portuguese Foundation for Science and Technology: PTDC/SAU-ENB/112294/2009, PEst-OE/SAU/UI0645/2014.

## Competing Interests

The authors declare there are no competing interests.

## Author Contributions

- João Rodrigues conceived and designed the experiments, performed the experiments, analyzed the data, wrote the paper, prepared figures and/or tables, reviewed drafts of the paper.
- Alexandre Andrade conceived and designed the experiments, analyzed the data, wrote the paper, reviewed drafts of the paper.

## Supplemental Information

Supplemental information for this article can be found online at http://dx.doi.org/10.7717/peerj.923#supplemental-information.

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
