# Peer review of "Synthetic neuronal datasets for benchmarking directed functional connectivity metrics"

_PeerJ, doi:10.7717/peerj.923_

## Round 0.1 · original submission · Major Revisions

The two reviewers differ in length and major conclusion.

If you take the comments from reviewer #2 as a first guide to rearrange the manuscript, you should also indirectly address the concerns of reviewer #1 about clarity.

Please make sure you also follow the various other detailed suggestions of reviewer #1. This should be mostly straightforward. Still due to the many points it seems a substantial task.

That reviewer #1 addresses the point that from his view the results mostly are trivial is not a critical point. However, you may want to argue from your side, which are the central results with most impact to the reader, which are more to be expected. Here also the suggestion from second reviewer #2 seems helpful:

"The comparison of the four models would benefit from a more detailed comparison. What are the benefits and disadvantages of each model as Granger Causality benchmarks? Which model(s) would the authors recommend to use as a benchmark model?"

Such a detailed comparison would clearly add to the value of the manuscript.

Reviewer 1 ·

Basic reporting

In the present manuscript the authors discuss the amount of causality in small network motifs of coupled neuronal oscillators of various type. These include autoregressive models, nonlinear stochastic differential equations, phase oscillators, and so-called Izhikevich columns. Based on simulated time series, they use different models to infer EEG and fMRI BOLD data. The causality is quantified by means of a spectral decomposition of Granger causality. In terms of parameter dependencies, the authors focus on coupling delays and coupling strengths. Their main finding is that causal structures persists - though to different extent – in all models and that the amount of causality can be used for benchmarking.

I found the manuscript not very accessible and the results mostly trivial. My judgement is based on the reasoning provided in the sections "Experimental Design" and "Validity of the Findings".

Experimental design

The inaccessibility of the manuscript is due to low-quality of the presentation as detailed in the following:

1. It is unclear why the particular neuronal models are chosen. Furthermore, the choice of local parameters is not well motivated. Do the results hold for other parameter values as well? For example, what is the effect of the noise terms? Concerning the influence of noise, Fig. 1 is misleading, because it suggests that the noise is added after down sampling although it is an ingredients of the local dynamics.

2. All model equations are introduced in a general notion as if the authors intended to simulate large networks. The simulated coupling topologies presented in the Results section are rather small and involve only a few elements. A schematic figure, which illustrates these coupling structures, would have been helpful. In addition, the relevance of the considered network motifs is unclear.

3. The introduction of the various models are incomplete and sometimes confusing. For instance, the delayed interaction matrix D (line 82) is missing in line 88, the exact form of the sigmoid S in Eqs. (1) and (2) is omitted, the different effects of excitatory and inhibitory neurons on the synaptic input is not reflected in Eq. (5) (line 176), the scale factor s (line 185) is not used in Eqs. (6), etc. Some equations appear wrong - or at least incomplete - such as Eq.(8) (line 281, do the authors meant to write |H|?) or line 310 (missing brackets in the denominator). Some variables are used multiple times such as S (lines 127 and 228).

4. Overall, the manuscript appears to be sloppily written. Abbreviations are used before they are introduced, e.g. AR, TR; others introduced twice, e.g. DFT (lines 65 and 113); some abbreviations are non-standard, e.g. blood activation level dependent (BOLD) or not used later in the text, e.g. dDTF or PDC. Information on the parameter selection is redundant and repeated in the Results section although they are already given in the Methods section. Some internal references are wrong such as Fig.1 (lines 215 and 227), which should refer to Fig.2. In the caption of that figure, not all symbols are explained. The spelling is inconsistent, e.g. non-linear or nonlinear, delayed-coupled or delay-coupled, and the wording could be improved, e.g. “in the 33 Hz” (lines 308 and 312) or “avoid initial conditions” (Results section), which probably means “to remove transient effects”. Speaking of initial conditions, it is unclear from which values the different simulations are started. This is of particular importance for systems involving delays. The parameter intervals used in the simulations are given in a non-standard fashion, e.g. “[4, 2:20:100] milliseconds” (line 336), which appears to be a programming-language-specific syntax. I assume that the above example should indicate 100 values equally spaced in the interval from 4.2 ms to 20 ms. There are many out-of-line symbols, in particular in the Results section.

Validity of the findings

The finding that increasing coupling strengths lead to stronger causal influence seems obvious. In addition, the independence with respect to delay times in the coupling is not discussed and might be due to different timescales of the local dynamics in the compound system. Investigating dynamics of coupled elements, the authors should have also considered other measures of synchrony (order parameter, Pearson correlation etc.).

Additional comments

To summarize, the results of this manuscript appear trivial and the presentation should be improved in various parts in order to make the investigation more accessible and to allow reproduction of the findings. In the present form, the manuscript is not suitable for publication.

Reviewer 2 ·

Basic reporting

No comments

Experimental design

No comments

Validity of the findings

No comments

Additional comments

This article compares four different generative models that can serve as benchmarks. However, the implications of the comparison is not clear and some methods need a further explanation.


Major comments:
- The ‘results and discussion’ sections contains a high level of detail and novel equations. Some of this content would be better placed within the Methods section.
- The comparison of the four models would benefit from a more detailed comparison. What are the benefits and disadvantages of each model as Granger Causality benchmarks? Which model(s) would the authors recommend to use as a benchmark model? For this, the authors could add a new schematic figure that provides an overview of the performance of each of the models.
- From the title it is currently unclear what connectivity the authors mean: directed connectivity could, for example, also stand for partially unidirectional structural fiber tract connectivity. Possibly ‘directed effective connectivity’ would be more precise?

Minor comments:
- line 29: define AR
- line 65: define GC
- Figure 1: explain the definition of TR and Fs in the legend
- equation (4): remove (t) from the equation (see Izhikevich, 2003 article)
- line 172: there are several potential behaviours of pyramidal cells. Is the described one of the type ‘regular spiking’? The authors should mention the types for both excitatory and inhibitory cells.

---

## Round 0.2 · Minor Revisions

I'm happy to inform you that only minor revisions are required.

Reviewer 1 ·

Basic reporting

In their revision, the authors improved the quality of presentation to large extent; mainly by re-organization of the manuscript and corrections to the text. For example, it is now explicitly stated that the study considers a network motif consisting of two nodes. I am thankful for the detailed point-by-point response by the authors, which clarifies the intent of the work, and I feel that the purpose of the study is now better accessible.

Experimental design

The introduction of the models has been greatly improved and the considered models are now well described in the Methods section. I also appreciate the summary of the findings in new Table I, which presents all results in a concise way.

Some minor comments, which could be easily implemented:

1. The abbreviation “BOLD” should always be used as “BOLD signal” or similar (abstract, introduction, and conclusion).

2. The notation of the MVAR model should be revised: the index j runs from 0 to p-1 in line 92 and from 1 to p in line 99; x_n should be replaced by X_n for consistency with Eq.(1) in line 99; the index j in phi_j should be written as a double index nj in lines 99 and 101.

3. I am still unsure that I fully understand the notation of the parameter ranges. Take, for instance, d_21 in line 138: Figure 4 indicates a range of [4ms ,100ms], but it is not clear how many values were used, i.e. what the increment of d_21 is. I suggest to clarify this point in the Methods section.

4. Related to the previous point: Is the scale of the y-axis (from 4s to 100s) in Figs.4D and 7D correct? Shouldn't it rather be milliseconds as stated in line 273?

5. There a still some typos: “non-linear” in the abstract; “time series” in the captions of Figs. 6, 7, 9, and 10.

Validity of the findings

The findings appear to be valid and the main result is that all four considered models are able to generate causal interactions. This holds for analysis of the simulated data and the inferred EEG and BOLD signals. I understand that the exact dependence (or independence) on the system parameters is of secondary importance and that the validity with respect to larger networks requires more work in future investigations.

Additional comments

In summary, there is still room for further improvement in the presentation (see section Experimental Design below), after which the text should be accepted for publication.

Reviewer 2 ·

Basic reporting

No Comments

Experimental design

No Comments

Validity of the findings

No Comments

Additional comments

The authors have successfully addressed all comments.

---

## Round 0.3 · accepted · Accept

The reviewer agrees that you have carefully taken all points into account. Thus, I'm happy to inform you that your manuscript is accepted.

Reviewer 1 ·

Basic reporting

The authors have carefully taken into account all points raised in my previous report. They removed the technical issues, further improved the notation, and corrected typos. Since my earlier criticism was mainly related to these technicalities and the authors substantially improved the accessibility of the manuscript already in their first revision by highlighting the scope of the study, it is my opinion that the paper is now suitable for publication in PeerJ.

Experimental design

All points of the previous report have been sufficiently addressed.

Validity of the findings

No further comments. See previous report.

Additional comments

I have no further comments and recommend acceptance. See “Basic Reporting”.